# Detection of Brain-Derived Cell-Free DNA in Plasma

**DOI:** 10.3390/diagnostics14222541

**Published:** 2024-11-13

**Authors:** Camilla Pellegrini, Francesco Ravaioli, Sara De Fanti, Chiara Pirazzini, Chiara D’Silva, Paolo Garagnani, Claudio Franceschi, Francesca Bonifazi, Pier Luigi Zinzani, Massimiliano Bonafè, Maria Guarino, Raffaele Lodi, Pietro Cortelli, Caterina Tonon, Micaela Mitolo, Luisa Sambati, Luca Morandi, Maria Giulia Bacalini

**Affiliations:** 1IRCCS Istituto delle Scienze Neurologiche di Bologna, 40139 Bologna, Italy; camilla.pellegrini@ausl.bologna.it (C.P.); francesco.ravaioli@ausl.bologna.it (F.R.); sara.defanti@ausl.bologna.it (S.D.F.); maria.guarino@isnb.it (M.G.); raffaele.lodi@isnb.it (R.L.); pietro.cortelli@isnb.it (P.C.); caterina.tonon@isnb.it (C.T.); luisa.sambati@isnb.it (L.S.); luca.morandi2@unibo.it (L.M.); 2Department of Medical and Surgical Sciences—DIMEC, University of Bologna, 40126 Bologna, Italy; chiara.pirazzini5@unibo.it (C.P.); paolo.garagnani2@unibo.it (P.G.); pierluigi.zinzani@unibo.it (P.L.Z.); massimiliano.bonafe@unibo.it (M.B.); 3Department of Pharmacy and Biotechnology (FaBiT), University of Bologna, 40126 Bologna, Italy; chiara.dsilva2@unibo.it; 4IRCCS Azienda Ospedaliero-Universitaria di Bologna, 40138 Bologna, Italy; francesca.bonifazi@unibo.it; 5Laboratory of Systems Medicine of Healthy Aging, Institute of Biology and Biomedicine and Institute of Information Technology, Mathematics and Mechanics, Department of Applied Mathematics, Lobachevsky State University, 603950 Nizhny Novgorod, Russia; claudio.franceschi@unibo.it; 6Istituto di Ematologia “Seràgnoli”, IRCCS Azienda Ospedaliero-Universitaria di Bologna, 40138 Bologna, Italy; 7Department of Biomedical and Neuromotor Sciences, University of Bologna, 40138 Bologna, Italy; 8Department of Medicine and Surgery, University of Parma, 43126 Parma, Italy; micaela.mitolo@unipr.it

**Keywords:** cfDNA, DNA methylation, target bisulfite sequencing, epihaplotypes, neurodegeneration

## Abstract

**Background**: Neuronal loss is a major pathological feature of neurodegenerative diseases. The analysis of plasma cell-free DNA (cfDNA) is an emerging approach to track cell death events in a minimally invasive way and from inaccessible areas of the body, such as the brain. Previous studies showed that DNA methylation (DNAm) profiles can be used to map the tissue of origin of cfDNA and to identify molecules released from the brain upon cell death. The aim of the present study is to contribute to this research field, presenting the development and validation of an assay for the detection of brain-derived cfDNA (bcfDNA). **Methods**: To identify CpG sites with brain-specific DNAm, we compared brain and non-brain tissues for their chromatin state profiles and genome-wide DNAm data, available in public datasets. The selected target genomic regions were experimentally validated by bisulfite sequencing on DNA extracted from 44 different autoptic tissues, including multiple brain regions. Sequencing data were analysed to identify brain-specific epihaplotypes. The developed assay was tested in plasma cfDNA from patients with immune effector cell-associated neurotoxicity syndrome (ICANS) following chimeric antigen receptor T (CAR-T) therapy. **Results**: We validated five genomic regions with brain-specific DNAm (four hypomethylated and one hypermethylated in the brain). DNAm analysis of the selected genomic regions in plasma samples from CAR-T patients revealed higher levels of bcfDNA in participants with ongoing neurotoxicity syndrome. **Conclusions**: We developed an assay for the analysis of bcfDNA in plasma. The assay is a promising tool for the early detection of neuronal loss in neurodegenerative diseases.

## 1. Introduction

Liquid biopsy is an innovative diagnostic method based on the analysis of circulating cells and biomolecules released by diseased tissue into accessible biofluids, such as blood and its derivatives [1]. While, in principle, the term liquid biopsy can be applied to any type of biomolecule (proteins, metabolites, and nucleic acids), it is most often used to indicate the analysis of circulating nucleic acids. Circulating cell-free DNA (cfDNA) molecules are double-stranded fragments of endogenous DNA released by cells into the bloodstream and other body fluids. The typical size of cfDNA is ~167 bp, corresponding to the length of DNA wrapped around the nucleosome core and protected from DNase cleavage. cfDNA molecules can be released upon active and passive mechanisms in both physiological (e.g., aging) and pathological (e.g., cancer) conditions [2]. In healthy individuals, cfDNA levels tend to be low, between 0 and 100 ng/mL, but pathological conditions involving cell damage and death can boost the release of cfDNA from the affected tissue into the bloodstream.

Both next-generation sequencing (NGS) and non-NGS methods, such as droplet digital PCR, can be applied to the study of cfDNA [3]. These approaches can be used to identify alterations in the cfDNA sequence (for example, the presence of somatic mutations and rearrangements), fragmentation pattern, and epigenetic modifications (DNA methylation and hydroxymethylation) associated to a certain disease. In addition, the epigenetic features of cfDNA molecules can be exploited to develop tissue-specific biomarkers of cell death [4]. It is well known that the specification of tissue identity during development largely relies on the establishment of definite epigenetic patterns (including nucleosome positioning, histone modifications, and DNA methylation) at certain genomic regions. Assuming that they are stable during life and that they are not altered by diseases, the identification of these epigenetic patterns in cfDNA fragments can be used to trace back the tissue-of-origin of the molecules and, as a consequence, to estimate the contribution of a certain tissue to the total cfDNA pool.

Among the various epigenetic modifications that can be harboured by cfDNA molecules, DNA methylation (DNAm) is probably the most suitable for the development of low-cost, streamlined, and targeted analytical methods. DNAm consists in the covalent binding of a methyl group to 5′ position of a cytosine, usually in a cytosine−guanine (CpG) context. Several methods to measure DNAm at targeted, genome-wide, and whole-genome levels have been developed [5], allowing the generation of large DNAm datasets from different human tissues and cell types. These datasets provide DNAm atlases that can be used to identify genomic regions with tissue-specific DNAm profiles [6,7], which in turn can be analysed in cfDNA molecules to track their tissue-of-origin. Lehmann-Werman et al. were among the first authors to use this approach, developing target bisulfite sequencing assays to quantify the proportion of plasma cfDNA molecules having a certain tissue-specific DNAm profile [8].

Since then, the use of DNAm-based liquid biopsy to detect cell death events in the brain has become a viable option. In their original study, Lehmann-Werman et al. developed two assays to detect brain-derived cfDNA (bcfDNA) by bisulfite sequencing [8]. The first one, targeting two genomic regions with a DNAm profile specific for oligodendrocytes, was applied to plasma cfDNA obtained from patients with multiple sclerosis. Almost contextually, Olsen et al. presented an alternative protocol to quantify oligodendrocyte-derived cfDNA in multiple sclerosis patients based on the analysis of DNAm of a different genomic region, namely the myelin oligodendrocyte glycoprotein (MOG) gene [9]. The second assay was developed to include a genomic region with a DNAm profile that more comprehensively distinguished brain cells from the other tissues (in other words, not specific for a certain brain cell type) and was applied to participants with ischemic brain damage after cardiac arrest and with traumatic brain injury [8]. In subsequent works, these assays were progressively updated, including multiple genomic regions, generic for the brain or specific for neurons, oligodendrocytes, and astrocytes [10,11]. In parallel, other groups developed alternative assays based on the bisulfite sequencing of other regions with brain-, neuron-, or glia-specific DNAm profiles and applied them to cfDNA isolated from patients with mild trauma [12], mild cognitive impairment (MCI), or Alzheimer’s disease (AD) [13].

The objective of the present study is to further contribute to the development of a liquid biopsy assay for the detection of bcfDNA. We selected a set of genomic regions with brain-specific DNAm profiles and experimentally validated them in a large dataset of DNA extracted from different human tissues. Finally, we tested the developed assay in samples from patients with immune effector cell-associated neurotoxicity syndrome (ICANS) following chimeric antigen receptor T (CAR-T) therapy, as this disease may cause the release of brain DNA in whole blood. An overview of the study flow is depicted in Figure 1.

## 2. Materials and Methods

### 2.1. Publicly Available Datasets

The Core 15-state model is a classification system defining 15 possible chromatin states predicted on the basis of 5 histone modifications that was developed by NIH Roadmap Epigenomics Consortium. Chromatin status data according to the Core 15-state model was downloaded from the Epigenomics Roadmap website (https://egg2.wustl.edu/roadmap/web_portal/chr_state_learning.html#core_15state, accessed on 16 March 2021) for the following adult human tissues and primary cell types—E067: brain angular gyrus; E068: brain anterior caudate; E069: brain cingulate gyrus; E071: brain hippocampus middle; E072: brain inferior temporal lobe; E073: brain dorsolateral prefrontal cortex; E074: brain substantia nigra; E027: breast myoepithelial cells; E029: CD14 primary cells; E032: CD19 primary cells; E034: CD3 primary cells; E046: CD56 primary cells; E051: CD34 primary cells; E062: peripheral blood mononuclear primary cells; E063: adipose nuclei; E065: aorta; E066: liver; E075: colonic mucosa; E076: colon smooth muscle; E077: duodenum mucosa; E078: duodenum smooth muscle; E079: oesophagus; E087: pancreatic islets; E094: stomach; E095: heart left ventricle; E096: lung; E097: ovary; E098: pancreas; E100: psoas muscle; E101, E102: rectal mucosa;E103: rectal smooth muscle; E104: heart right atrium; E105: heart right ventricle; E106: sigmoid colon; E107: skeletal muscle; E108: skeletal muscle; E109: small intestine; E110: stomach mucosa; E111: stomach smooth muscle; E112: thymus; and E113: spleen. The Bedops software (v2.4.39) [14] was used to overlap the Illumina Infinium 450 k probes and the Epigenomics Roadmap chromatin states.

A total of 415 human DNase-seq datasets were downloaded from the ENCODE repository on 14 December 2021: 100 datasets from healthy nervous tissues (including Ammon’s horn, caudate nucleus, head of caudate nucleus, cerebellar cortex, cerebellum, frontal cortex, globus pallidus, inferior parietal cortex, myocardium inferior, myocardium superior, medulla oblungata, midbrain, middle frontal area, middle frontal gyrus, occipital lob, pons, posterior cingulate gyrus, putamen, sciatic nerve, spinal cord, superior temporal gyrus, and tibial nerve); 199 from healthy non-nervous tissues (including adrenal gland, aorta, pancreas, breast tissue, cardiac septum, oesophagus, coronary artery, chorion, colon, eye, femur, gastrocnemius, heart, forelimb, hindlimb, kidney, liver, lung, skin, adipose tissue, gallbladder, placenta, prostate gland, muscle tissue, retina, spleen, stomach, testis, thyroid gland, tongue, umbilical cord, ureter, urinary bladder, uterus, and vagina); 46 from brain tissues of Alzheimer’s disease patients; 8 from brain tissues of patients with cognitive impairment; and 48 from brain tissues of patients with mild cognitive impairment condition.

The *GEOmetadb* Bioconductor package (v1.67.0) [15] was used to interrogate the Gene Expression Omnibus (GEO) repository [16] on 4 November 2021, using the following search terms: “GPL13534” to include only datasets based on the Illumina Infinium HumanMethylation450 BeadChip, and “control”, “normal”, “non-tumour”, and “health” to select datasets including healthy subjects. We excluded datasets generated on whole blood, as they were highly over-represented, but we retained the datasets generated on isolated blood cell types (PBMCs, CD4 cells, CD8 cells, and neutrophils). The resulting list was manually revised, and it is reported in Appendix A. DNAm levels of brain and non-brain tissues were compared using *limma R* package (v3.54.2), a statistical approach to identify differentially methylated targets using linear models with an empirical Bayes method [17]. Differential analysis was performed only on CpG probes for which DNAm values were available in more than 10% of the investigated datasets (1378 out of 1740 probes); CpG probes with a Benjamini–Hochberg-adjusted *p*-value < 0.01 were retained as statistically significant.

### 2.2. Biological Samples

The Genotype-Tissue Expression (GTEx) DNA samples analysed in this manuscript were obtained from 44 different autoptic tissues collected from 4 subjects—2 males and 2 females (Appendix A).

The plasma samples were collected in the framework of a study on patients affected by relapsed/refractory B-cell lymphoma after CD19 CAR T-cell therapy [18,19]. Participants were enrolled at IRCCS Azienda Ospedaliero—Universitaria di Bologna after the signature of a written informed consent form. The study was approved by the local ethics committee, conducted according to the Helsinki declaration, and registered at ClinicalTrials.gov (NCT04892433, NCT05807789). The 5 samples analysed in this manuscript were all collected 3–5 days after CD19 CAR T-cell infusion. Three participants had ICANS at the time of collection, while two participants did not have ICANS at the time of collection and did not develop it later. Blood samples were collected by venipuncture in EDTA tubes and centrifuged at 1500 g for 15 min at room temperature to separate plasma. Plasma aliquots were stored at −80 °C until cfDNA extraction.

cfDNA was extracted from 1 mL of plasma using QIAamp MinElute ccfDNA Mini Kit (Qiagen, Hilden, Germany) and quantified with Qubit dsDNA HS Assay Kit and Qubit Fluorimeter 3.0 (Thermo Fisher Scientific, Waltham, MA, USA). cfDNA length and quality were verified by the 5200 Fragment Analyzer system (Agilent, Santa Clara, CA, USA).

### 2.3. Target Bisulfite Sequencing

For the analysis of GTEx samples, 200 ng of genomic DNA were bisulfite-converted using the EZ DNA Methylation Kit (Zymo Research, Irvine, CA, USA). For the analysis of cfDNA from plasma, 4–10 ng was bisulfite-converted using the EZ DNA Methylation-Direct Kit (Zymo Research, Irvine, CA, USA). A two-step PCR was used to perform target bisulfite sequencing. 

For GTEx samples, a PCR was performed for each target in a final volume of 5 μL using Phusion U (Thermo Fisher Scientific, Waltham, MA, USA) added with 1 M Betaine (Thermo Fisher Scientific, Waltham, MA, USA), 0.3 μM of forward and reverse primers, 1.75 mM MgCl_2_ (Thermo Fisher Scientific, Waltham, MA, USA), 200 μM dNTP (Thermo Fisher Scientific, Waltham, MA, USA), and 5 ng of bisulfite-converted DNA. Primer sequences are reported in Appendix A. Thermal cycler conditions were set as follows: 1× cycle at 95 °C for 1′40″; 1× cycle at 98 °C for 1′; 1× cycle at 58 °C for 2′; 1× cycle at 72 °C for 1′; 36 cycles at 98 °C for 10″, 58 °C for 40″, and 72 °C for 20″; 1× cycle at 72 °C for 5′; and hold at 4 °C. PCR products were pooled sample-wise and purified using 0.9X MagSi-NGS plus beads (MagTivio, Nuth, The Netherlands).

For the simultaneous detection of the selected targets on plasma cfDNA, we performed a multiplex PCR in a final volume of 50 μL containing 1X Phusion U Multiplex PCR Master Mix (Thermo Fisher Scientific, Waltham, MA, USA), 0.3 μM of forward and reverse primers, 1M Betaine (Thermo Fisher Scientific, Waltham, MA, USA), and 12.9 μL of bisulfite-converted cfDNA. Thermal cycler conditions were set as follows: 1× cycle at 98 °C for 1′; 36× cycles at 98 °C for 30″ (denaturation), 58 °C for 3′ (annealing), and 72 °C for 30″ (extension); 1× cycle at 72°C 5′; and hold at 4 °C. PCR products were purified using 0.9X MagSi-NGS plus beads (MagTivio, Nuth, The Netherlands). 

In the second PCR step, 7.5 μL of purified samples were indexed using Illumina Nextera XT Index (Illumina, San Diego, CA, USA) in a final volume of 50μL according to Illumina protocol. Indexed libraries were purified using 0.7X MagSi-NGS plus beads (MagTivio, Nuth, The Netherlands) and normalized to 4 nM. Sequencing was performed on an Illumina Miseq Platform with Micro v2 300PE chemistry (Illumina, San Diego, CA, USA). The expected read depth was 2000× for each genomic target for GTEx samples and 20,000× for each genomic target for cfDNA samples.

### 2.4. Data Analysis

Sequencing data were processed according to AmpliMethProfiler pipeline [20], as indicated in [21]. AmpliMethProfiler tool is able to assess the DNAm status of each CpG site within the target region at single DNA molecule level (1 = methylated, 0 = unmethylated, 2 = mismatch). In addition, AmpliMethProfiler returns the methylation status of adjacent CpG sites within the target at the single DNA molecule level (epihaplotypes). AmpliMethProfiler parameters were set as follows: *--primTresh* 0.8, *--threshLen* 0.2, *--dust* no. In the analysis on GTEX samples, all the tested targets had a mean read depth higher than 2000×, indicative of high sequence quality. For each target region, samples with a read depth < 500 were excluded from further analyses. The CpG dinucleotides residing within the PCR primers were trimmed. For each sample, epihaplotypes occurring only once and epihaplotypes containing mismatches were removed, and the frequency of each epihaplotype was calculated with respect to the total number of remaining reads. Heatmaps of epihaplotype frequencies were generated using the *heatmap.2* function from the *gplots* R library (v3.1.3.1). Clustering was performed using complete-linkage algorithm applied on Euclidean distances matrix (default *heatmap.2* settings). To evaluate the robustness and reliability of the inferred clusters, a multiscale bootstrap resampling (*n* = 2000) complete linkage clustering analysis was performed on Euclidean distance matrices using *pvclust* R package (v2.2-0). 

In addition, a parsimony analysis was performed. For every target, we considered each epihaplotype to be associated either to a “Brain” or “Other Tissues” signature, as described below. We calculated the percentage of brain samples in our GTEx cohort as 40/141 = 0.28 (Appendix A). For epihaplotypes that are less frequent in the brain compared to other tissues (median frequency in the brain ≤28th percentile), we assigned a “Brain” signature to samples whose epihaplotype frequency was lower than 28th percentile. Vice versa, for epihaplotypes which are more frequent in the brain (median frequency in the brain ≥28th percentile), we assigned a “Brain” signature to samples whose epihaplotype frequency was higher than 72nd percentile. Using this assumption, we built the parsimony tree for each target region. We then compared their topology with complete linkage clustering trees we had previously generated on Euclidean distance matrix. Parsimony phylogenetic trees were generated using *pratchet* function of *phangorn* R library (v2.11.1).

To calculate the fraction of bcfDNA in each sample, we used an approach similar to the one previously described [11,22], calculating the percentage of reads with a brain-specific epihaplotype for each target genomic region (all the CpGs methylated for APC2, all the CpGs unmethylated for the other target genomic regions), and summing them [22]. 

## 3. Results

### 3.1. Identification of Genomic Regions Harbouring Brain-Specific Epigenetic Marks

Similarly to other studies performed so far [7,8,12,13], we used genome-wide DNAm datasets generated with the Illumina Infinium microarrays to identify CpG sites harbouring brain-specific DNAm patterns. In order to improve the chance of selecting robust brain-specific regions, we pre-filtered the Illumina Infinium microarray probes on the basis of the histone modifications found in the corresponding genomic regions in different tissues. The rationale underlying this approach is that DNAm is not a standalone epigenetic mark, but it is strictly co-regulated with other regulatory mechanisms, including histone modifications [23]. Specifically, we used the Core 15-state model elaborated by the NIH Roadmap Epigenomics Consortium, and we compared the data from 7 brain regions with those from 34 non-brain tissues and primary cell lines (Section 2). This analysis resulted in 1740 probes mapping in genomic regions with a non-overlapping chromatin conformation status between brain and any other non-brain tissue (Appendix A).

We then mined the GEO repository for genome-wide DNAm data generated from healthy brain and other tissues using the Illumina Infinium HumanMethylation450 BeadChip. This search resulted in 95 datasets (Appendix A); 27 of them (for a total of 1693 samples) were generated from different brain regions (caudate nucleus, cerebellum, cingulate gyrus, entorhinal cortex, frontal cortex, temporal cortex, motor cortex, occipital cortex, sensory cortex, and hippocampus), and the remaining 1734 samples were generated from tissues other than brain (including CD4T cells, CD8 T cells, monocytes, neutrophils, PBMCs, adipose tissue, lung, aorta, breast, colon, endometrium, uterus, bone, kidney, liver, muscle, oral mucosa, ovary, pancreas, pituitary, prostate, skin, and thyroid). Of the 1740 CpG sites identified above, 1378 were available in more than 10% of the investigated datasets, allowing the comparison of their DNAm values between brain and non-brain tissues (Section 2; Appendix A). We identified 1318 probes significantly differentially methylated between brain and non-brain tissues (Benjamini–Hochberg-adjusted *p*-value < 0.01; Appendix A). To refine the list of potential targets, we applied additional selection criteria. First, we focused on the top 20 significantly hypermethylated and the top 20 hypomethylated probes (defined according to the difference in mean DNAm between brain and non-brain tissues) and manually selected those having consistently large DNAm differences between brain and non-brain tissues. We then selected probes mapping in genomic regions harbouring at least 2 CpG sites in the 140 bp upstream and downstream. Finally, we evaluated the corresponding genomic regions for their hypersensitivity to DNase, using ENCODE Regulation DNase HS data from brain and non-brain tissues. We prioritized CpG sites mapping in regions with low levels of DNase hypersensitivity in brain tissue, as this increases the likelihood that these genomic regions are not highly digested and are therefore detectable in the cfDNA pool. 

Collectively, these analyses resulted in the selection of five target CpG sites mapped in the following genes: *APC2* (hypermethylated in brain tissues), *PACRG* (hypomethylated in brain tissues), *FAM123* (hypomethylated in brain tissues), *DNER* (hypermethylated in brain tissues), and *C10orf90* (hypomethylated in brain tissues). Figure 2A–C and Appendix A report the DNAm values of the selected probes in brain and non-brain tissues from the GEO repository.

### 3.2. Experimental Validation by Targeted Bisulfite Sequencing in Different Human Tissues

To experimentally validate the brain-specific DNAm of the 5 CpG sites that we selected above, we used DNA samples from 44 autoptic tissues isolated from 4 healthy subjects, available in the framework of the Genotype-Tissue Expression (GTEx) project (Appendix A). Importantly, this collection included different brain regions for the same subject, including cortex, hippocampus, cerebellum, amygdala, caudate nucleus, nucleus accumbens, putamen, and substantia nigra.

The DNAm of the selected CpG sites and of the surrounding ones was measured by targeted bisulfite sequencing [24], considering an amplicon length of 170 bp at maximum, compatible with the average length of cfDNA fragments (Appendix A). For the APC2 target region, it was not possible to include cg10094078 in the PCR amplicon due to high CpG content, so we designed primers to cover CpGs immediately downstream this genomic region. We also applied the same validation strategy to three other target regions (encompassing the Illumina probes cg02619656, cg09787504, and cg23661000) that emerged in a recent study as having a brain-specific DNAm [10]. Of note, all those three probes were included in our list of 1318 CpG sites that were significantly differentially methylated between brain and non-brain tissues (Appendix A), although they did not rank among the top 20 hypo- or hypermethylated ones.

Figure 2D–F and Appendix A show the DNAm values of the selected CpG sites in the autoptic tissues from the GTEx collection. 

The differences between brain and other tissues were clearly confirmed for all the target CpG sites in the expected direction (hypermethylation for APC2 and DNER, hypomethylation for the other CpG sites). However, differences for C10orf90 and DNER were less striking, and a higher inter-tissue variability within both brain and non-brain was observed. These trends were also confirmed when considering the DNAm profile of all the CpG sites within the amplicon (Figure 2G–I and Appendix A). Interestingly, we found that some target regions showed notable differences in DNAm levels: FAM123, cg02619656, and cg23661000 had a stronger hypomethylation in cerebellum, while for PACRG, this structure had high DNAm levels, comparable to those of non-brain tissues.

### 3.3. Epihaplotype Analysis for Tissue Discrimination

We then analysed the validation dataset in order to quantify the frequency of the different epihaplotypes, i.e., the combination of methylated and unmethylated CpG sites along the same DNA molecule, in each sample (Appendix A). Previous studies suggested that this analytical approach increases the ability of discriminate between tissues. Accordingly, unsupervised clustering on epihaplotype frequencies clearly discriminated brain and non-brain tissues for all the amplicons (Figure 3 and Appendix A). We applied multiscale bootstrap resampling to confirm the ability of our targets to discriminate between brain and non-brain tissues. APC2, cg23661000, and FAM123 epihaplotypes were able to cluster brain tissues with the highest confidence (approximately unbiased *p*-value (AU) > 95%) (Appendix A). In addition, for each target region, we built a parsimony tree. The comparison of their topology with the result of hierarchical clustering further supports the ability of epihaplotypes to discriminate between brain and non-brain tissues (Appendix A).

For PACRG, FAM123, cg02619656, cg09787504, and cg23661000 target regions, the epihaplotype including all unmethylated CpG sites had a mean frequency of around 50% in brain tissues and a negligible frequency in all the other brain tissues. For the APC2 target region, epihaplotype frequencies were sparser; the profile with all CpG sites methylated reached the highest frequency, and two additional profiles harbouring only 1 unmethylated CpG site were enriched, although at a lower level, in brain compared to non-brain tissues. Finally, for C10ORF90 and DNER targets, there was a less marked difference in epihaplotype frequencies between brain and non-brain tissues and a higher variability among brain regions.

### 3.4. Validation of the Assay in Plasma Samples from Patients with Immune Effector Cell-Associated Neurotoxicity Syndrome

Similarly to the protocol previously described [11,22], we developed a multiplex assay to measure the percentage of plasma cfDNA molecules harbouring brain-specific DNAm epihaplotypes. On the basis of the results described above, we selected the best five out of eight of the tested targets: APC2, PACRG, FAM123, cg02619656, and cg09787504. C10orf90 and DNER were excluded because of their less marked differences in mean DNAm and epihaplotype frequencies between brain and non-brain tissues; FAM123, cg02619656, and cg23661000 had a similar DNAm profile (with a more marked hypomethylation in cerebellum), but the first two target regions were preferred because of the larger fold differences of the unmethylated epihaplotype between brain and non-brain tissues (Appendix A). We decided to maintain APC2 in the final assay because it was the only target hypermethylated in the brain compared to other tissues, despite the fact that the mean frequency of the fully methylated epihaplotype was relatively small. 

To explore the ability of the assay to quantify bcfDNA, we applied it to plasma cfDNA samples extracted from five patients affected by relapsed/refractory B-cell lymphoma who received CD19 CAR T-cell therapy. At the time of plasma collection, three patients were in the acute phase of ICANS, an adverse event in CD19 CAR T-cell therapy characterized by neurotoxicity and high levels of markers of neuroaxonal injury [25]. For this reason, we assumed that ICANS could be an appropriate clinical condition to validate our assay.

The multiplex assay successfully amplified the target loci with a high read depth, with a similar efficiency across samples (Figure 4A). All the three ICANS samples showed high levels of bcfDNA in comparison to patients without ICANS (Figure 4B), although the differences did not reach statistical significance (*t*-test *p*-value: 0.09), probably as a consequence of the small sample size. This result is in agreement with the neurotoxicity phenotype of ICANS patients and confirms the ability of the multiplex assay to detect bcfDNA in plasma.

## 4. Discussion

There is growing interest in the development of assays to quantify plasma cfDNA molecules deriving from the central nervous system that could be used as markers of brain damage and neurodegeneration [10,11,12,13]. Our study contributes to this research field, providing new insights on the selection and validation of genomic regions harbouring a brain-specific epigenetic profile.

Our selection strategy was based on the integration of chromatin-state maps from the Roadmap Epigenomics Project and of DNAm data from genome-wide DNAm datasets generated from different tissues. Almost all (1318 out of 1378) of the probes that we selected as having brain-specific chromatin states also had significantly different DNAm values between brain and non-brain tissues, confirming the appropriateness of this approach. It is worth noting that the large majority of the datasets that we used for the CpG sites selection are based on bulk tissues containing different cell types, both for brain (neurons, glia, endothelial, and ependymal cells) and for the other tissues. The recent generation of a methylation atlas from sorted primary cells [6] will allow to further refine the selection of genomic regions with cell-specific epigenetic patterns.

The large majority (1143 out of 1378, 83%) of brain-specific sites were hypomethylated in respect to the other tissues, consistent with previous results on tissue-specific differentially methylated regions (DMRs) [26,27]. The hypo- or hypermethylated status of DNA is an important aspect to be considered when choosing genomic regions to be analysed in cfDNA. There is indeed a close connection between DNAm status, nucleosome positioning, and DNA fragmentation by DNases, which ultimately results in the representation of cfDNA sequences released in circulation. In particular, genomic regions with increased DNAm levels tend to be more represented in the population of cfDNA fragments [28]. Previous studies on bcfDNA used opposite approaches; some assays focused on regions hypermethylated in the brain, as they are more likely to be present in the cfDNA pool [12], while others included only genomic regions that are hypomethylated in the brain, which tend to show more significant DNAm differences [10,13]. Here, we preferred to consider for validation both brain hyper- and hypomethylated genomic regions, checking available DNase-seq datasets from different tissues in order to avoid hypersensitive sequences that are less likely to be represented in the cfDNA pool. This process led to the selection of five Illumina probes, three hypomethylated (mapping in PACRG, FAM123, and C10orf90), and two hypermethylated (mapping in APC2 and DNER). Of note, APC2 target regions were also included in the assay developed by Chatterton et al. [12].

Another important aspect in the selection of genomic sequences with brain-specific methylation is their possible epigenetic modulation under physiological and pathological conditions. For example, the DNAm of genomic targets for bcfDNA detection should not be affected by sex and should not be modified during aging or in diseases, with particular regard to neurological diseases. As a proof of concept, we considered the results of a recent meta-analysis that we performed on epigenome-wide data, in which we reported Illumina probes that were associated to age, sex, or Alzheimer’s disease in different brain regions. None of the five selected probes was included in the lists of differentially methylated CpG sites, and the same applies to the three CpG probes (cg02619656, cg09787504, and cg23661000) from the assay by Lubotzky et al. [10] that we included in the validation steps.

We validated the brain-specific DNAm profile of the eight target regions in a large collection of DNA samples from several tissues, provided by the GTEx Biobank. To the best of our knowledge, this is the largest validation dataset used in similar studies so far. Furthermore, and more importantly, GTEx collection includes multiple brain regions, allowing the checking of epigenetic differences among them. We found that hypomethylation of FAM123, cg02619656, and cg23661000 was more pronounced in the cerebellum compared to other brain regions, and that, conversely, PACRG was hypomethylated in all the brain regions except for the cerebellum, which displayed DNAm values comparable to non-brain tissues. Considering that we analysed DNA extracted from bulk tissues, these differences can be due to the variable proportions of neuronal and non-neuronal cell types, as they harbour specific DNAm patterns [29,30,31]. However, we did not observe marked changes in DNAm values of the selected loci in the GEO datasets generated from sorted glia (GSE41826, GSE74486). An additional, not necessarily alternative, hypothesis is that the DNAm profile of the selected loci contributes to the functional identity of neuron subtypes and brain regions [32,33,34,35]. Future studies should exploit DNAm differences across different brain regions in order to develop bcfDNA assays for specific brain regions relevant for certain diseases, as discussed below.

To validate in vivo the ability of the multiplex assay to detect bcfDNA in plasma, we applied it to samples from patients who received CD19 CAR T-cell therapy, three of whom had ICANS at the time of sample collection. Previous studies showed that plasma levels of NfL, a marker of neuroaxonal injury, increase after infusion of CAR T-cell therapy and that they positively correlate with neurotoxicity severity [25]. Consistently, we found an increase of bcfDNA levels in ICANS patients, indicating that our assay is able to detect ongoing neuronal damage.

The detection of bcfDNA in plasma is of great interest in the field of brain diseases [36,37], but it is still in its infancy as biomarker when considering the indications of the FDA-NIH Biomarker Working Group [38]. The assays described so far have been tested in patients with traumatic brain injury, ischemic brain damage, multiple sclerosis, and psychosis [8,10,12]. As in our manuscript, the focus of these studies was more on the demonstration of the technical feasibility of the assays than on the evaluation of their potential as diagnostic biomarkers. Recently, Pollard et al. showed that neuron-derived cfDNA is increased in plasma from patients with AD. Even more interestingly, the authors reported higher levels of neuron-derived cfDNA in plasma from patients with mild cognitive impairment (MCI) who later progressed to AD, but not from stable MCI patients. To quantify neuron-derived cfDNA, in the study, the authors measured the DNAm of a single target genomic region surrounding an Illumina probe that was not among the 1740 CpG sites that we selected in our pipeline. Although preliminary, these results pave the way for the application of bcfDNA assays as prognostic biomarkers for the prediction of conversion from preclinical and prodromal stages to overt disease. In addition, detection of bcfDNA in plasma holds great potential as a monitoring biomarker for neurological diseases, given its accessibility, minimal invasiveness, and low cost that allow the assay to be serially repeated.

So far, most attempts to develop blood tests for neurological and neurodegenerative diseases have focused on the analysis of plasma proteins measured by ultra-sensitive methods [39,40]. One of the most promising protein candidates is NfL, whose blood levels increase in response to axonal injury and neurodegeneration [41,42]. However, higher NfL levels cannot be attributed to damage in a certain brain area or subpopulation of neurons, as it is ubiquitously expressed in neurons across the brain. Conversely, differential DNAm has been reported across the brain, and some genomic regions have a DNAm profile unique to a specific brain area [35]. The possibility of developing bcfDNA assays including these genomic regions is particularly attractive, as it would allow to identify the brain area in which neurodegeneration is occurring and to follow the spread of the disease through the brain [43,44]. Other important, albeit largely unexplored, aspects are related to the triggers of brain cell death in neurodegenerative diseases [45] and to the timing of cfDNA and protein release during disease onset and progression. For example, using in vitro and in vivo models, Pai et al. suggested that the release of cfDNA by amyloid-β (Aβ) occurs in the early phases of the disease, prior to the formation of Aβ plaques [46]. On the other side, as previously underscored, cfDNA has a half-life definitely lower than plasma proteins, which possibly makes it more useful for a real-time monitoring of neurodegeneration, but the factors that regulate its accumulation and clearance are largely unknown [12,37]. In addition, the effects of potential confounding factors on bcfDNA levels, such as blood–brain barrier permeability and kidney function, should be taken into account. Finally, it should be noted that other types of liquid biopsy biomolecules, such as metabolites and cell-free RNA [47,48,49], are under investigation as biomarkers of neurodegenerative diseases. It is likely that, in the future, the integration of different circulating biomolecules will be a beneficial strategy to improve protein-based biomarkers of neurodegeneration.

Our study has some limitations. First of all, as our validation dataset included only bulk tissues, we are unable to experimentally assess whether the DNAm values and epihaplotypes of the selected target regions differ between neurons and glial cells. Secondly, validation on cfDNA samples extracted from plasma was performed in a small number of samples from a single clinical setting in which we reasonably expected high levels of neuroaxonal injury markers. We are aware that these results may not be generalized to ICANS condition and that the higher levels of bcfDNA that we observed may be influenced by additional demographic and clinical confounding factors. Further ad hoc studies in a larger cohort should better explore these aspects. On the other side, our study has the merit of using a large DNA collection, including several tissues and brain regions, to analyse multiple genomic regions with potential brain-specific DNAm that were selected from a systematic analysis of publicly available datasets and from the recent literature.

In conclusion, we developed and validated an assay for the analysis of bcfDNA in plasma, which holds promise for the early detection and monitoring of neuronal loss in neurodegenerative diseases and in other clinical conditions characterized by neuronal damage.

## Figures and Tables

**Figure 1 diagnostics-14-02541-f001:**
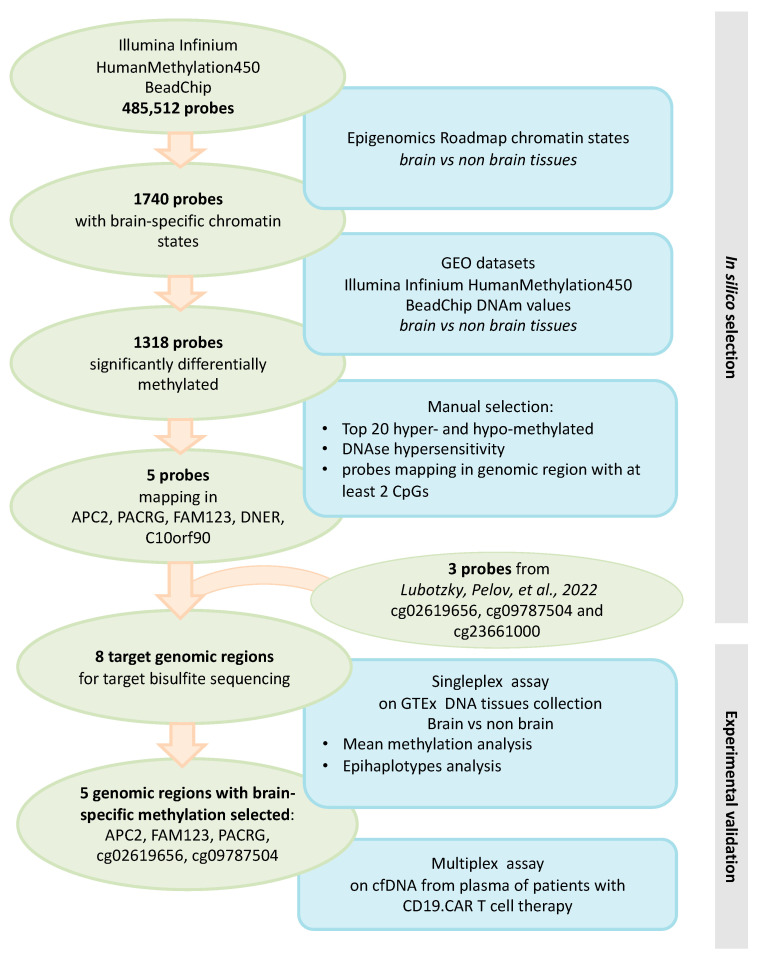
Study design. The flowchart summarizes the design of the study, including the selection and validation of genomic regions with brain-specific DNAm [10].

**Figure 2 diagnostics-14-02541-f002:**
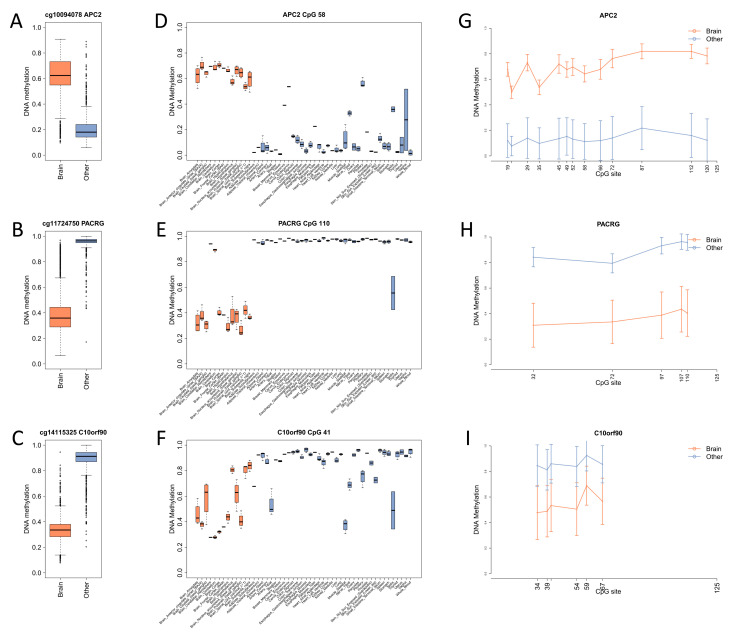
DNAm profile of brain-specific targets. (**A**–**C**) The boxplots report the DNAm values of 3 representative microarray probes resulting from the comparison between brain and non-brain (other) tissues from the GEO repository. (**D**–**F**) The boxplots report the DNAm values of the 3 CpG sites depicted in panels (**A**–**C**), measured by target bisulfite sequencing in 44 GTEx brain and non-brain tissues. (**G**–**I**) The line plots show the mean DNAm levels of the CpG sites included in the 3 representative target regions in brain and non-brain tissues (other). Note that the figure shows 3 out 8 of the selected targets; the remaining targets are reported in Appendix A.

**Figure 3 diagnostics-14-02541-f003:**
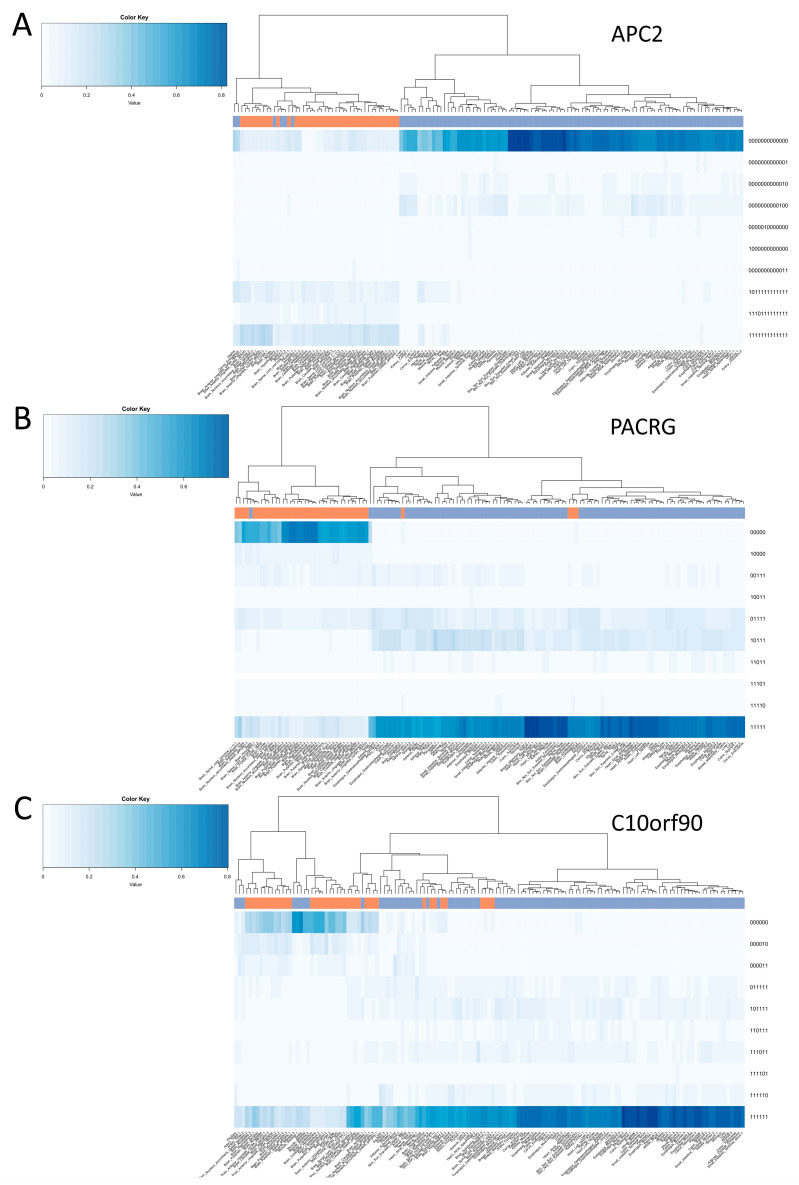
Epihaplotype frequencies of brain-specific targets. The heatmaps represent the frequencies of epihaplotypes (rows) in the GTEx samples (columns) for targets (**A**) APC2, (**B**) PACRG and (**C**) C10orf90. Heatmap colours go from white (frequency of a certain epihaplotype near to 0% in a certain sample) to dark blue (frequency of a certain epihaplotype near to 100% in a certain sample). The colour bar above the heatmap indicates brain (orange) and non-brain (blue) tissues. For the sake of clarity, the heatmaps include only the 10 most frequent epihaplotypes for each target region. Dendrograms were generated using complete-linkage clustering on Euclidean distances matrix. Note that the figure shows 3 out 8 of the selected targets; the remaining targets are reported in Appendix A.

**Figure 4 diagnostics-14-02541-f004:**
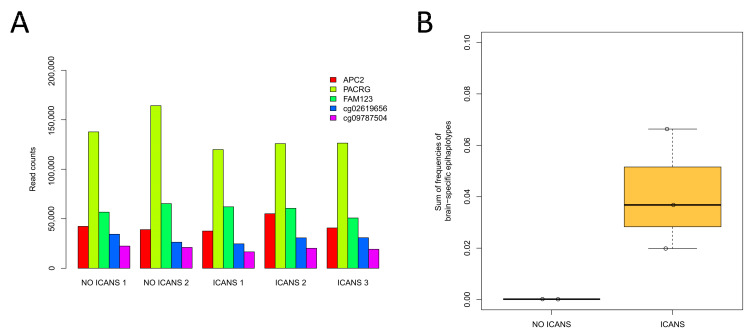
Analysis of bcfDNA in plasma samples from patients with and without ICANS. (**A**) Read depth of the 5 target regions analysed through target bisulfite sequencing in the plasma samples from the 5 patients who received CD19 CAR T-cell therapy. (**B**) Levels of bcfDNA in CAR T patients who developed or did not develop ICANS.

## Data Availability

The original sequencing data presented in the study are openly available in the NCBI Sequence Repository Archive (SRA) at PRJNA1157590. Bioinformatic workflows can be downloaded at https://github.com/LabBrainAgeing/Brain-cell_free_DNA_Analysis.

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
