# Peer review of "Detection of Brain-Derived Cell-Free DNA in Plasma"

_diagnostics, 2024, doi:10.3390/diagnostics14222541_

Round 1
Reviewer 1 Report
Comments and Suggestions for Authors
The manuscript presents a relevant study utilizing recent advances in the detection of brain-derived cell-free DNA (bcfDNA) through DNA methylation assays. This technology holds significant potential for early diagnostic applications in neurodegenerative diseases and immune-related neurotoxicity. Although the results are promising, the study has some methodological limitations, particularly in the validation stage, with a relatively small sample size (N), which impacts the broader applicability of the findings. Despite this, the manuscript is well-structured, addresses a clear research question, and provides robust and pertinent conclusions. Below are the major and minor comments addressing methodological clarity and scientific rigor.
Major Comments:
- In the figures presenting heatmaps (Figure 3 and S3), the methods used for clustering and visual representation are not explicitly described. It is important for readers to understand the methodology underlying the cluster formation to evaluate the robustness of the findings. Could the authors specify in the methodology and figures the exact clustering algorithms used (e.g., hierarchical clustering, k-means)? Additionally, was any distance metric (e.g., Euclidean, Manhattan) applied to assess the similarity between samples?
- Another critical point is the validation of these clusters. Clustering results, especially in genomic datasets, can vary depending on algorithmic parameters. Have the authors performed any bootstrap resampling (e.g., with r=1000 or more) to ensure the stability and reproducibility of the clustering? If not, it is recommended that they perform this analysis to confirm that the observed clusters are not artifacts of random variation. Similarly, a parsimony analysis should be conducted to assess if simpler models could yield similar clustering, thereby validating the complexity of the chosen method. This would strengthen the confidence in the final results and clustering interpretation.
- While the manuscript focuses on the detection of bcfDNA in plasma, there is limited discussion on how these findings could correlate with clinical progression in patients over time. For neurodegenerative diseases, longitudinal data are often crucial for understanding whether early biomarkers can predict clinical outcomes. Could the authors provide further insights into the potential of this assay for ongoing clinical monitoring, rather than just static diagnostic evaluation.
Minor Comments:
- The manuscript notes that CpG sites were selected for bisulfite sequencing. However, there is no detailed explanation of the criteria for CpG site selection. Were only CpG sites with significant brain-specific methylation considered? Were any regions excluded due to poor sequencing quality or other factors? Clarifying this would provide a better understanding of the assay’s precision.
- The study includes patients with ICANS as one of the disease cohorts. However, the potential confounding factors specific to this group, such as concurrent therapies or inflammatory status, are not well addressed. Could the authors provide more information on how they controlled for possible confounders in ICANS patients, which may have impacted cfDNA levels?
- As noted, the sample size for this validation study is relatively small, which limits the generalizability of the conclusions. Could the authors comment on how this limitation affects their findings? Are there any plans to validate the assay in a larger cohort, and how might this change the current conclusions?
Overall, the manuscript is written in clear and comprehensible scientific English. However, there are a few areas where minor corrections in language usage would improve clarity. Specifically:
- In the introduction, sentences that describe the DNA methylation assay development could be more concise.
- A few grammatical inconsistencies in subject-verb agreement were noted, particularly in sections describing the methodology.
A thorough copy-editing pass is recommended before final submission.
Reviewer 2 Report
Comments and Suggestions for Authors
Detection of brain-derived cell free DNA in plasma
1. Need a brief explanation of cfDNA (circulating free DNA) for readers who might be unfamiliar with the term. This will ensure clarity for a broader audience.
2. In liquid biopsy, several types of biomolecules are used; this study is using cell-free DNA (cfDNA). It is important for this study to emphasize the aspects that make cfDNA superior to other biomolecules. Doing this will increase understanding of this study.
3. In the Introduction section, various techniques for cfDNA sequencing and analysis of fragmentation patterns are mentioned. It may be appropriate and helpful to provide the some names of the specific techniques used for these analyses.
4. Use simple terminology for terms such as "the core 15-state model", as this may be difficult for unfamiliar readers to understand.
5. There is a need to further clarify the details of statistical methods such as the limma R package, so that interested researchers could repeat the analysis.
6. What does "1378 of 1740 probes" mean in line 152? A brief description of the probes filtering criteria would be helpful.
7. For plasma samples, consider explaining the importance of using CD19.CAR T cell patients following relapsed/refractory B cell lymphoma. This may include why these samples are valuable for study, particularly in the context of CAR T-cell therapy and potential links to ICANS.
8. Due to the low image quality of Figure 2 and Figure 3, it is quite difficult to read any details associated with them. The author can improve the quality of this image if possible.
9. In the manuscript, the authors use 'circulating cfDNA' in some places and 'cfDNA' in others. It would be appropriate to use one word consistently.
10. In the manuscript, references are appended directly to the end of the line, as seen in line 90. The entire manuscript needs to be checked.
Comments on the Quality of English LanguageMinor editing of English language required.
Round 2
Reviewer 2 Report
Comments and Suggestions for Authors
The authors addressed all my concern, therefore, I would like to recommend this manuscript for publication in journal.